# Particle Swarm Optimization Algorithm for Guided Waves Based Damage Localization Using Fiber Bragg Grating Sensors in Remote Configuration

**DOI:** 10.3390/s22166000

**Published:** 2022-08-11

**Authors:** Rohan Soman, Alex Boyer, Jee Myung Kim, Kara Peters

**Affiliations:** 1Institute of Fluid Flow Machinery, Polish Academy of Sciences, Fiszera 14, 80-231 Gdansk, Poland; 2Department of Mechanical and Aerospace Engineering, North Carolina State University, Campus Box 7910, Raleigh, NC 27695, USA

**Keywords:** guided waves, fiber Bragg grating (FBG) sensors, remote bonding, damage localization, particle swarm optimization

## Abstract

Structural health monitoring (SHM) systems may allow a reduction in maintenance costs and extend the lifetime of the structure. As a result, they are of interest to the research community. Ideally, the SHM methods should be low cost, while being able to detect and localize small levels of damage reliably and accurately. The fiber Bragg grating (FBG) sensors are light in weight, insensitive to electric and magnetic fields, and can be embedded. The edge filtering configuration for transduction allows the use of FBG for guided wave (GW) sensing. This sensitivity may be further enhanced through their application in the remote bonded configuration. This paper provides a proof-of-concept for the use of remotely bonded FBG for damage localization. In order to improve the computational efficiency, a particle swarm optimization (PSO) based algorithm is developed. The PSO allows a significant improvement in the computation time which makes it better suited for real-time damage localization. The proposed objective function is based on the exponential elliptical approach. First, the suitability of the PSO for damage localization is shown. Then the performance of the chosen objective function is compared with the brute-force algorithm as well as other objective functions found in the literature. The methodology is employed on a simple aluminum plate. The results indicate that indeed the objective function along with the PSO is suitable for damage localization. Also as the objective function is developed taking into consideration the specific challenges with the use of FBG sensors, performs better than the other objective functions as well as the brute force algorithm.

## 1. Introduction

Structural deterioration occurs during the course of the structure’s lifespan. Detection of this deterioration at an early stage is invaluable for maintenance planning. Furthermore, this may allow an extension of the lifetime, which in turn reduces life-cycle costs. Thus detection of the deterioration has attracted interest in the field of structural health monitoring (SHM). A range of techniques making use of different damage-sensitive parameters such as vibration-based SHM [1], strain-based SHM [2], guided wave-based SHM [3] etc. have been developed.

GW-based techniques have been popularly employed for plate-like structures, as they offer the possibility to reliably detect and localize the damage. Several different sensors such as the lead zirconate titanate (PZT), macro fiber composites, optical fiber sensors, acoustic non-contact transducers, as well as a laser Doppler vibrometer, have been used for the sensing. The optical fiber sensors such as the fiber Bragg grating (FBG) sensors and Fabry Perot interferometers offer low-weight alternatives for sensing [4,5]. The FBG sensors also offer several other advantages such as immunity to magnetic and electric fields, small size, ability to be multiplexed etc.

FBG sensors were traditionally used for strain and temperature measurement in the wavelength division multiplexing (WDM) approach wherein the change in the reflected wavelength is used to determine the measure. The sensitivity of the FBG sensors in the WDM configuration is not sufficient for GW measurements. So they are used in the edge filtering configuration. The edge filtering approach has been discussed in detail by Soman et al. [6].

In addition to the edge filtering approach, the sensitivity of the FBG sensors can be further enhanced by employing it in the remote bonding configuration. The remote bonding configuration is shown in Figure 1a. In the remote bonding configuration the Symmetric (S) and Antisymmetric (A) modes encounter the bond with the optical fiber and get converted into optical fiber longitudinal (L) and flexural (F) modes. The conversion of plate modes in the optical fiber is shown in Figure 1b. The phenomenon is discussed in detail in [6,7]. The optical fiber acts as an excellent wave guide for the mechanical waves and these fiber modes travel along the fiber with minimal attenuation. Along the fiber, as the FBG is in free condition (without constraints), it undergoes greater deformation leading to a higher sensitivity. In the directly bonded configuration, a shear lag effect is observed which results in a lower response in the directly bonded FBG than in the remotely bonded FBG. The phenomenon was studied in detail by Huang et al. [8]. Wee et al. quantified the improved sensitivity and reported that the remote bonding configuration yields an 81% increase in sensitivity for some conditions [9].

Some researchers have leveraged the added sensitivity of the FBG sensors in the remote configuration for damage detection [9,10]. Yu et al. [11,12] proposed the use of FBG in remote configuration for sensing in extremely high temperatures. The remote FBG may be paired up with non-contact excitation techniques and actuation and sensing may be achieved in spite of the high temperature of the structure to be monitored. The same non-contact actuation was paired up with a remote FBG in an improved adhesion configuration [13]. The remote FBG is located equidistant from two perpendicular bonds on the same fiber. The two bonds allow overcoming of the directional sensitivity of the FBG sensors. The full wavefield information is then used for detecting damage in the structure. In both the described studies, the full wave field measurements utilized make the approach time-consuming and not suitable for in-service SHM. In the studies using sparse arrays of sensors or actuators, work by Wee et al. [14] which uses the remote bonded FBG for self-referencing stands out. They use two bonds equidistant from the remote FBG on either side. The interference of the waves coupled at the two bonds is then used for detecting damage. This method is limited only to damage detection in a limited region of the structure. For damage localization, Yu et al. [11]. used the difference in the arrivals of the A0 and S0 wave from an acoustic event. This allowed determination of the distance between the acoustic event and the bond location, but is not useful for knowing the damage location precisely. Hence based on the literature review, there is no work where the remotely bonded FBGs are used for damage localization. This paper addresses this lack of research in the area and presents a proof of concept for the use of FBG sensors in the remote configuration for the damage localization.

The work builds on the work by Soman et al. [15,16]. who used the ellipse mapping algorithm for damage detection and localization using directly bonded FBG sensors. Soman et al. [15,16] discretized the plate into smaller pixels and calculated the damage index (DI) for all the pixels using a brute force method. This technique leads to additional computational costs and delays in the measurements and the determination of the condition of the structure. In order to reduce the computational load. Balasubramaniam et al. [17]. proposed a multi-level approach where the regions of the damage were identified in the first step and the damage localization was carried out only in the identified region. This approach indeed reduces the computational load but is not always possible as it needs a large number of actuator-sensor pairs. The use of additional actuators or sensors increases the cost significantly and hence is not desirable and may not be possible on large structures. Hence this paper proposes a damage location algorithm based on particle swarm optimization (PSO). The damage localization problem is posed as an optimization problem with the objective to maximize the damage index. The paper was inspired by similar work by Chen et al. [18,19] where the evolutionary algorithm was used for damage localization. The key difference with the above work is the development of a different objective function, which for the current problem using FBG sensors in remote configuration performs better than the objective function developed by Chen et al. [19].

The paper provides a proof of concept for the use of FBG sensors in the remote bonding configuration for damage localization. A PSO-based algorithm is developed to determine the damage location while reducing the computational load as compared to the damage mapping technique. The paper also presents a new objective function for damage localization and compares it with a metric previously developed in the literature. The validation is performed on a simple aluminum plate with simulated damage. The presented results indicate that indeed the PSO-based algorithm works efficiently and successfully localizes damage with reduced computation time. The reduction in computation time makes the method more suitable for real-time SHM.

The rest of the article is organized as follows: the next section highlights the methodology for damage localization including the main features of the PSO, the objective functions, and their implementation. The Section 3 outlines the setup used for the validation of the methodology. The Section 4 presents the results for the damage localization along with some sensitivity and comparative studies. The last section draws some conclusions based on the presented results and identifies areas for further research.

## 2. Methodology

Damage localization through an ellipse imaging method can be done with a pixelated approach, where a damage index is calculated on every discrete pixel on a grid using the Equation (Equation 1) [20]. This approach is termed the brute-force approach.
(1)$$D\left(x,y\right)=\sum_{i=1}^{N_{p}}\exp\left(-\frac{\left|\Delta T\right|}{\tau }\right)$$
where D(x,y) is the damage index at a particular pixel, $N_{p}$ is the number of actuator-sensor paths, ΔT is the timing difference between the measured and expected time-of-flight (TOF), and τ is a sensitivity factor. This approach has been shown to be effective but is computationally expensive and time consuming.

In order to develop a computationally efficient approach, damage localization is treated as a function optimization problem. As a result, the damage may be localized in fewer damage index (DI) calculations. This method involves creating an objective function that returns the likelihood of a particular location being the damaged location, and using existing mathematical optimization tools to find the location that results in the highest probability.

Particle swarm optimization (PSO) is a common optimization technique that is by design simple and computationally efficient. PSO involves a group, or swarm, of candidate solutions, known as particles, that have a position and velocity [21]. Each particle will move a small amount every iteration, attracted to both its individual best position it has found so far, and the overall best position in the swarm. The attraction to each particle’s individual best is intended to cause particles to find and fully explore the search space, and the attraction to the global best is intended to achieve convergence. This tendency for each particle to interact with its own previous locations and the best performing location gives PSO algorithms an element of swarm intelligence not found in techniques like genetic algorithms or gradient descent. For a 2D localization problem, the PSO is ideally suited and has fewer artificial constraints. This allows a more thorough search of the possible solution space and efficiency in the implementation. Hence the PSO was chosen as the algorithm for the optimization in this paper.

### 2.1. Particle Swarm Optimization and Its Implementation

The basic procedure for a PSO algorithm is as follows. The flowchart is provided in Figure 2.

1. Initialize Swarm: A population of particles called a swarm is created, with positions randomly distributed uniformly throughout the search space. The particle’s fitness is found by passing its position into the objective function up for optimization. In our research, standard elliptical imaging TOF methods are used to find ellipses focused on each actuator-sensor (AS) pair, and the fitness is based on the particle’s proximity to these ellipses which corresponds to the likelihood that the particle’s location is the damaged location. The particle with the highest global fitness is found and its location recorded.

2. Update Particle Positions: The change in a particle’s position from one iteration to the next is known as the particle’s velocity. Each particle’s velocity is determined at each iteration, and then added to the particle’s position to get the particle’s next position. Each particle’s velocity is determined as a weighted vector sum of 3 vectors: the particle’s previous velocity, the displacement vector to the location with the highest fitness that particle has found so far, and the displacement to the location with the highest global fitness found so far. The weight given to its previous velocity is known as its momentum coefficient, and the weights given to the individual maxima and global maxima are known as acceleration coefficients. Adjusting these acceleration coefficients changes each particle’s attraction to its individual and global maxima. Note that on the first iteration, as each particle’s individual maxima is its current position, the particle with the highest fitness does not move.

3. Evaluate Fitness: The fitness is evaluated for the new position of each particle. The fitness value is then used to determine the global maximum and the individual maximum which are then used to check for convergence as well as determination of the weighting coefficients. The fitness function used in this research is discussed in more detail later.

4. Check Terminating Condition: If the maximum global fitness value exceeds a certain criterion, or if the average particle velocity is below a certain threshold, or if the maximum number of iterations has been reached, the algorithm returns the position with the best fitness found as the optimal solution.

The number of particles, also known as swarm size, along with the momentum and acceleration coefficients, are input factors that need to be tuned for best performance.

### 2.2. Objective Function

The objective function is based on the candidate location’s proximity to ellipses focused on each actuator-sensor pair. The measured time of flight (TOF) is compared with the expected time, and the difference is passed through a negative exponential function as shown in Equation (Equation 1). To combat the less than perfect overlap of ellipses from separate actuator-sensor (AS) pairs, this value is calculated for the particle’s position as well as a few positions nearby. These values are averaged to provide the final particle fitness. This helps to smooth the steep slope of the negative exponential scaling function when the timing difference is very small, however, it comes at the cost of a more computationally expensive objective function, which usually must be counteracted by using fewer particles.

In addition to the above-mentioned objective function, one other objective function was studied. The objective function is based on the work by Chen et al. [19] who used the number of ellipses passing through a point on the structure as an objective function. They applied this metric on a sensor network consisting of 6 actuator sensors, leading to 15 AS pairs. The suitability of the metric for the damage localization in our case is studied and compared with the exponential metric.

### 2.3. Experimental Setup

The experimental validation was conducted on a 61 cm × 60.5 cm × 0.08 cm 6061 aluminum plate, with weights added to simulate damage. The additional weights offer a repeatable and reversible way of introducing wave scatterers in the structure and are commonly used. The added mass is a scatterer for the waves and is considered equivalent to drilled holes [23,24]. The weights used were rare-earth magnets on either side of the plate to ensure firm attachment. The magnets were 2 cm in diameter, and weighed a total of 25 g. Three piezoelectric transducers (PZT) for wave excitation were bonded to the plate using cyanoacrylate adhesive. The PZTs, driven by an arbitrary waveform generator (Agilent 33500B Series) and voltage amplifier (Krohn-Hite Model 7602M Wideband) set to an output voltage of 60 V, generated bursts of a 300 kHz excitation signal to generate $S_{0}$ and $A_{0}$ Lamb waves. The excitation signal was a 5.5 wavelength Hanning windowed sine wave.

3 polyimide-coated optical fibers were bonded radially to the first PZT at a distance of 16 cm in accordance with the procedure by Wee et al. [9], with fiber Bragg grating (FBG) sensors in the remotely bonded configuration in each fiber. The distance from the fiber bonding point to the FBG is different for each fiber, and are 31 cm, 71 cm, and 80.25 cm respectively. The ends of each fiber were placed in index matching gel to reduce reflections from the end of the fiber. A point to note is that the aim of the experiments was to provide a proof-of-concept of the method. The deployment of the number of actuators, sensors, and their location was based on prior experience of the authors. In reality, the deployment of the actuators and sensors needs to be optimized to ensure reliable performance. Several techniques tackling this problem may be found in the literature [15,24,25]. The optimization of the network is identified as an area of future work.

A tunable laser source (Photonetics/NetTest TUNICS-Plus) is used for generating the laser wavelength at the midpoint of the rising edge of the FBGs, and the signals reflected from the FBG were captured using a circulator, photo-detector (Newport 1544-A Near-IR Fiber-Optic Receiver), and oscilloscope (Keysight DSO5032A 5000 Series). The photo of the setup is shown in Figure 3 and the schematic of the experimental setup is shown in Figure 4.

For each of the 7 damage cases as well as the healthy condition, the FBG sensor response is recorded for each actuator-sensor pair, with only one actuator active at a time. In order to remove any biases or high-frequency noise, a bandpass filter in the range of 150 kHz to 450 kHz was applied to the raw data. The filtered signals were then used for further processing. For each case the residual signal, which is the difference between the healthy case and each damaged case is found. The Hilbert envelope is fitted on the residual and smoothed. The envelope and smoothing allow us to overcome minor misalignment in the healthy and damaged signal. This leads to a more repeatable and robust identification of the peaks. The peaks in the smoothed and enveloped residual are identified using the ’findpeaks’ command in MATLAB. These peaks are used in the elliptical fitness function described previously.

A point to note is that the optical fibers show directional sensitivity, when the excitation is in an axial direction to the fiber, coupling of the A and S modes is more dominant. The domination of the coupling of S and A modes lasts until around 75°. When the actuation is more than 75° or near perpendicular to the adhesive bond the coupling of the SH waves into the optical fiber is more dominant. When the SH waves are dominant, the identification of the direct arrival of the A0 wave is difficult. This makes signal processing more challenging. So, only the AS pairs that lead to A and S mode coupling are chosen for the signal processing namely A1 with all three Bonds (Bond 1, Bond 2, and Bond 3), A2 with Bond 3, and A3 with Bond 1. The A0 mode is chosen as it has been shown to be better for damage localization. The A0 wave has a smaller wavelength and hence may be used for detecting small damage.

## 3. Results and Discussion

### 3.1. Particle Swarm Optimization

The PSO was run with the chosen parameters for damage localization. Figure 5 shows the progression of the maximum fitness and the mean fitness for the entire population. Having a large swarm size allows a better global search, but increases the computation time. The choice of the other parameters affects the local search and the rate at which convergence occurs. In the case presented, by the 4th iteration, the global maxima identified is close to the identified damage location. The stopping criterion is met at iteration number 9.

The population progression for all iterations is shown in Figure 6. There is a clear convergence in the localized region of the damage localization. As can be seen, in Figure 6i, the simulated damage is at the location (28.9, 18.6) while the identified location is at (28.27, 17.78) which gives an error of 1.03 cm which is smaller than the 2 cm diameter of the mass used as damage. This shows that indeed the damage localization using PSO technique is suitable.

### 3.2. Damage Localization

Once the parameters for the PSO were established, it was used for damage localization of the other 6 damage scenarios, the identified locations have been shown in Figure 7. The error in the damage localization is shown in Table 1. Based on the small value of error, it can be concluded that indeed the damage localization is possible using the PSO algorithm.

### 3.3. Statistical Studies

The PSO is a meta-heuristic technique, as a result, there is no certainty that the PSO will indeed find the global maxima. The probability of reaching the global optima depends on the chosen coefficients of the PSO, the swarm size, and the cut-off criterion. To determine the suitability of the chosen parameters, the damage localization was run 100 times for the damage scenario 7. Figure 8 shows the identified damage for each run. As can be seen, only 4 of the runs yield a damage localization outside the acceptable zone (4 cm distance from actual damage). The acceptable zone is determined based on the diameter of the magnets, and the 5 μs tolerance given to the peak picking algorithm. The Figure 9 shows the error for each of the runs. It confirms that the error in localization is within acceptable limits for 96% of the runs. The statistical results from the PSO are in line with the expectations for a meta-heuristic method where the global search either yields the global maxima (true detection) or is stuck at a local maxima (false detection). The performance can be improved by fine-tuning the parameters of the optimization to ensure a better global search. This improved performance comes at the cost of additional computation time. The acceptable probability of detection for aerospace applications is around 10% [26,27,28] and hence the 96% achieved by the PSO is considered satisfactory.

### 3.4. Comparative Studies

In order to ascertain the better suitability of the proposed method, the objective function, and the performance were compared with two other studies. The comparative studies are presented here.

#### 3.4.1. Comparison with Brute Force Algorithm

The brute-force algorithm involves the computation of the damage index for each pixel along the structure. The pixel size of 0.5 cm × 0.5 cm has been used by the authors in their previous work and was used. The chosen pixel size for the plate leads to a 122 × 123 grid size. This requires a total of 15,006 computations of the DI. For the damage 7 scenario, the time needed for the brute force technique was 59.63 s. On the other hand, for the PSO in the same damage scenario, with a swarm size of 100, a total number of function evaluations at 5500, and required for an average of 18.5 s. Hence, the number of computations and the time required is approximately 1/3 rd. Figure 10 gives the DI plot for damage scenarios D1 and D4.

For scenario D1, there are 3 hot spots that are of high value and possible locations of damage. The real damage scenario can only be identified by applying appropriate thresholding to highlight the highest DI location. For, the D1 scenario, the highest DI is at location (28.5, 17.5) which gives an error of 1.17 cm which is comparable to the error obtained for the damage localization using the PSO technique. For the scenario D4, the highest DI is at location (19, 30.5) which gives the error of 3.99 cm. The error in localization is within an acceptable range, but is higher than the PSO-based technique. So, using the PSO technique, there is an obvious improvement in the time required for the damage localization. At the same time the accuracy of the localization is not affected.

#### 3.4.2. Comparison with Other Cost Functions

As mentioned before, PSO is a meta-heuristic technique, which depends on random population generation. Hence a comparison of the objectives based on their effectiveness using the PSO may be difficult. As a result the brute force technique (dividing the entire structure into pixels and calculating the cost function for each pixel) was used for the comparison. The plate was divided into 75 × 75 grid and the DI for both cost functions were calculated. The Figure 11 shows the DIs for two damage cases D2 and D3.

In Figure 11a,c it can be seen that the location with maximum DI is not unique. This is caused due to the relatively small resolution of values the cost function can take. The cost function can take only integer values and the maximum is capped by the number of AS pairs. On the other hand, for the DI based on an exponential, the DI can be any positive real number. Furthermore, it can be clearly seen that the maxima of the DI are only in one location which is close to the actual damage. Hence it can be concluded that the exponential-based cost function is more suitable for damage localization. It has to be conceded that the computation of the number of ellipses is way less demanding and requires 1/5th the time for the computation of the exponential function. This may be considered as a drawback of the exponential DI as the cost function.

## 4. Conclusions

GW-based SHM using FBG sensors is a field that is receiving renewed interest in the last few years. The paper provides a proof of concept of the use of FBG sensors in remote configuration for the localization of damage in an aluminum plate. To the best knowledge of the authors, this work is the first time remotely bonded FBG sensors have been used for damage localization. In order to reduce the time needed for the localization of the damage, a PSO-based algorithm has been implemented making use of an innovative objective function. The parameters for the PSO have been chosen based on engineering judgment to ensure a thorough global as well as local search. The PSO indeed allows the damage localization in significantly lower time as opposed to the brute force algorithm.

The performance of the proposed objective function is qualitatively and quantitatively better than the objective function proposed in the literature. The objective function based on the number of ellipses works well where a large number of AS pairs are present. As the FBG sensors are passive sensors, the number of AS pairs are severely reduced. Hence, the new objective function developed takes into consideration the passive nature of the FBG sensors and hence is more suitable for use with the FBG sensors. It should be conceded that the study is applied for the proof of concept on a simple aluminum plate, additional complexities such as anisotropy of the structure and the presence of more complex structural features need to be taken into account before the above technique may be used for damage localization in real structures and is identified as an area of future work. Although the proposed cost function yields the correct damage location, there is a chance that the PSO gets stuck in local maxima, in order to avoid such scenarios, a multi-objective optimization needs to be developed and is planned for future work. 

## Figures and Tables

**Figure 1 sensors-22-06000-f001:**
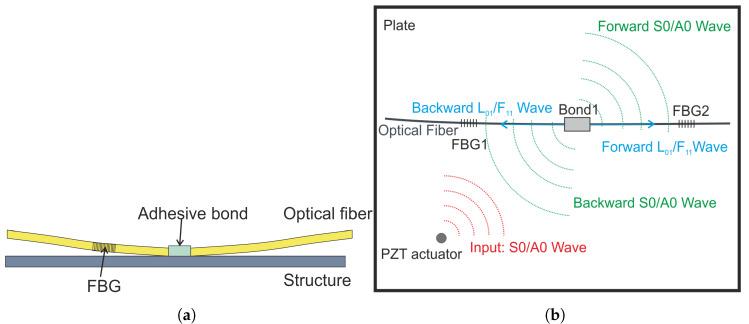
(**a**) Remote bonding configuration (**b**) Mode conversion in remotely bonded FBG.

**Figure 2 sensors-22-06000-f002:**
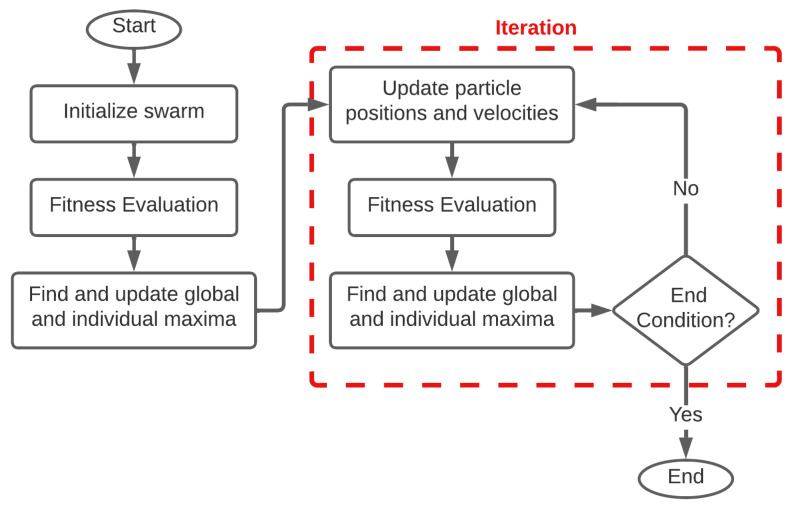
Particle swarm Optimization Flow Chart based on [22].

**Figure 3 sensors-22-06000-f003:**
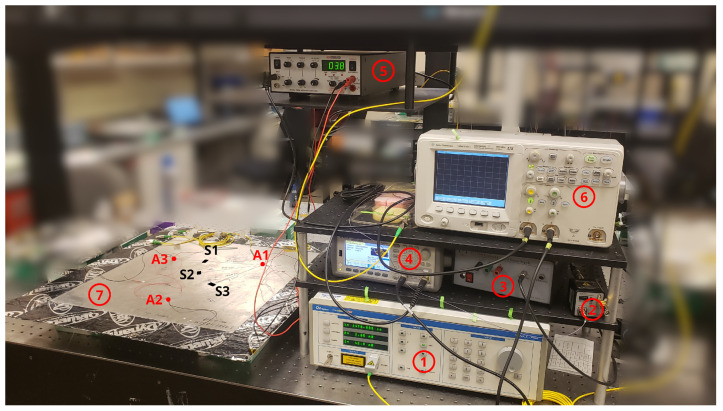
Experiment Setup. 1—Tunable Laser, 2—Photodetector, 3—Power Supply, 4—Waveform Generator, 5—Voltage Amplifier, 6—Oscilloscope, 7—Experiment Sample. 8—Actuators (A1, A2, A3), 9—Sensors (S1, S2, S3).

**Figure 4 sensors-22-06000-f004:**
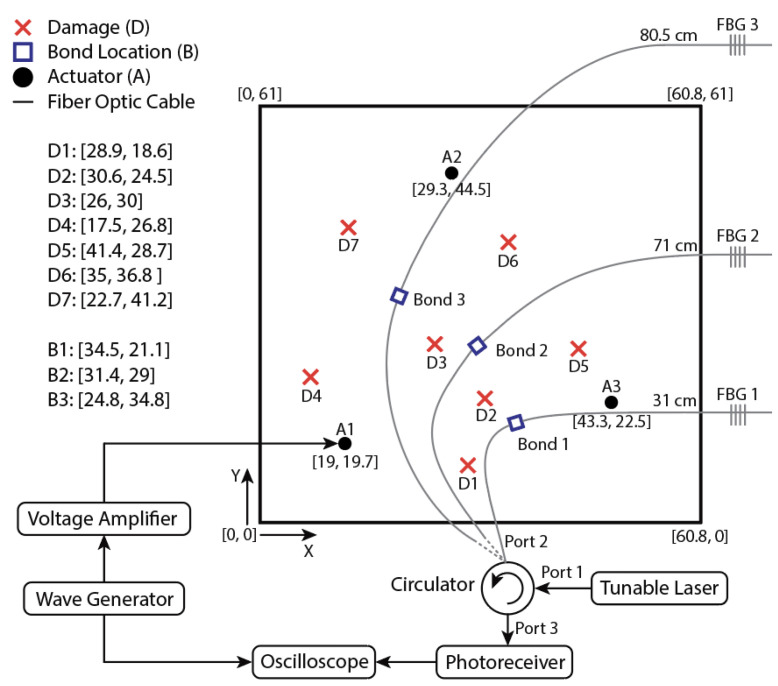
Schematic of the experimental setup (Coordinates are in cm).

**Figure 5 sensors-22-06000-f005:**
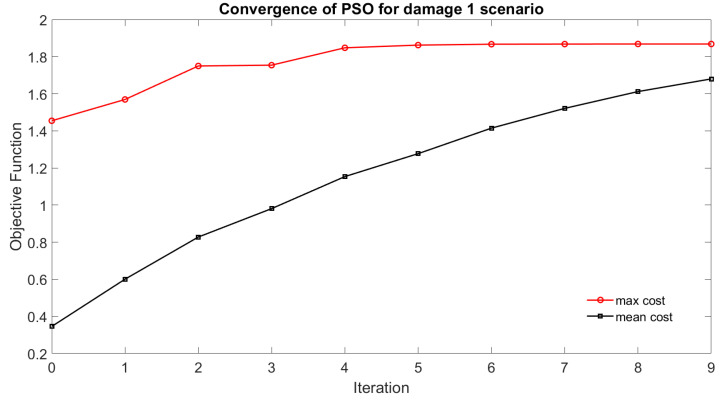
Convergence of the PSO to global maxima.

**Figure 6 sensors-22-06000-f006:**
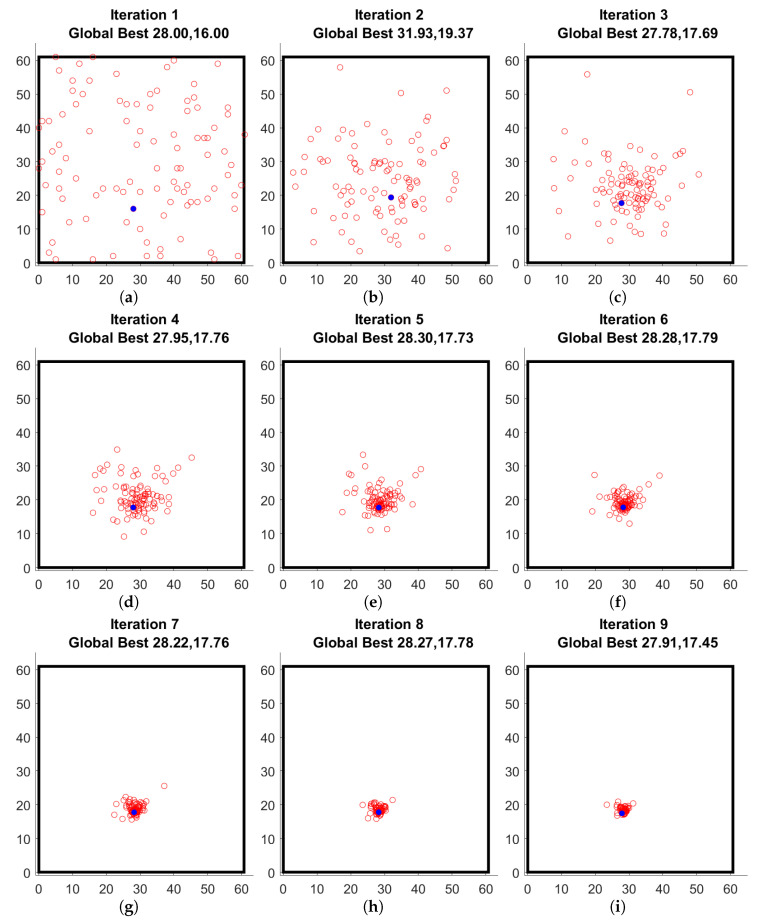
Swarm locations at each iteration (global best marked with a filled blue circle).

**Figure 7 sensors-22-06000-f007:**
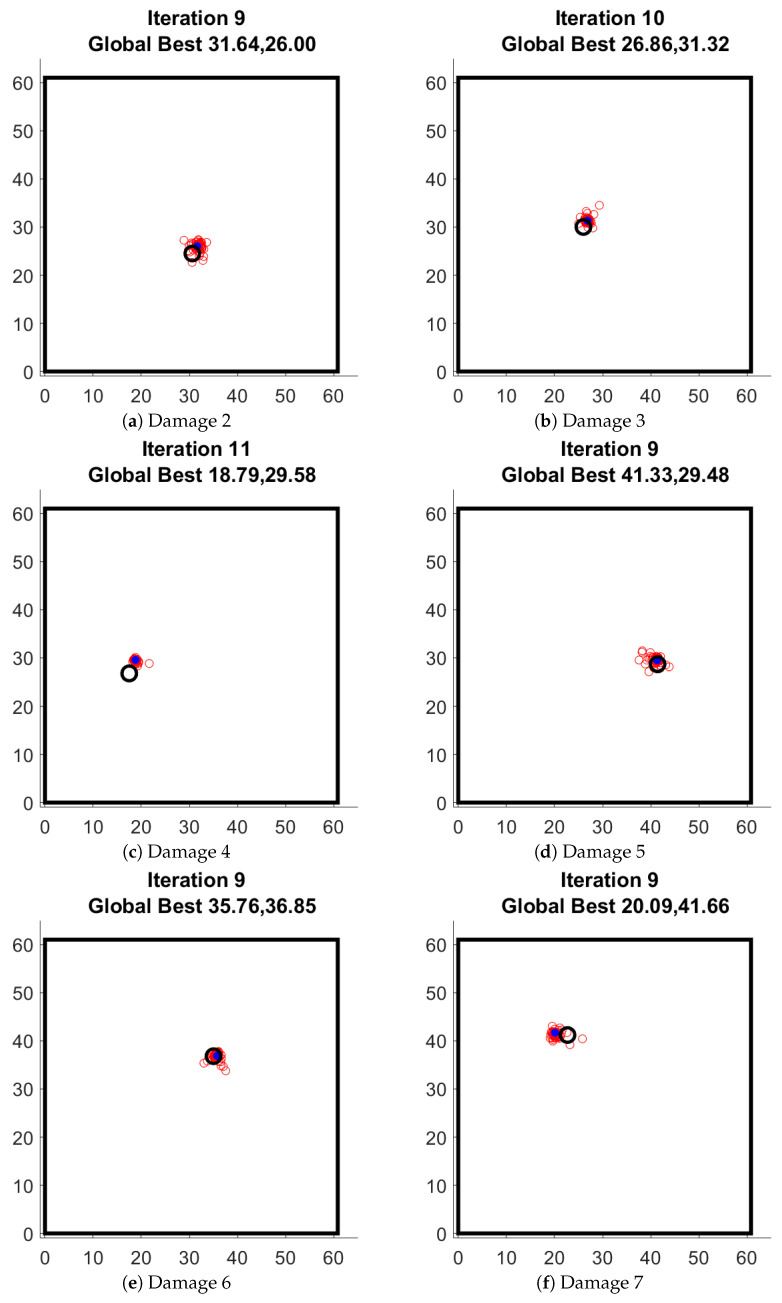
Damage localization for 6 damage scenarios. (black circle shows the real location of magnet).

**Figure 8 sensors-22-06000-f008:**
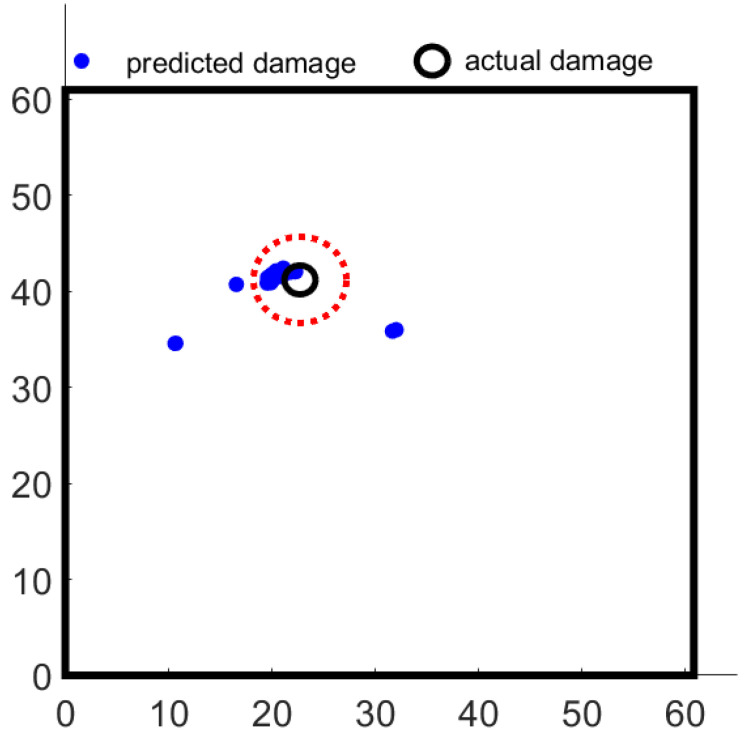
Damage localization for damage 7 for 100 PSO runs (red dotted circle indicates the acceptable region for localization).

**Figure 9 sensors-22-06000-f009:**
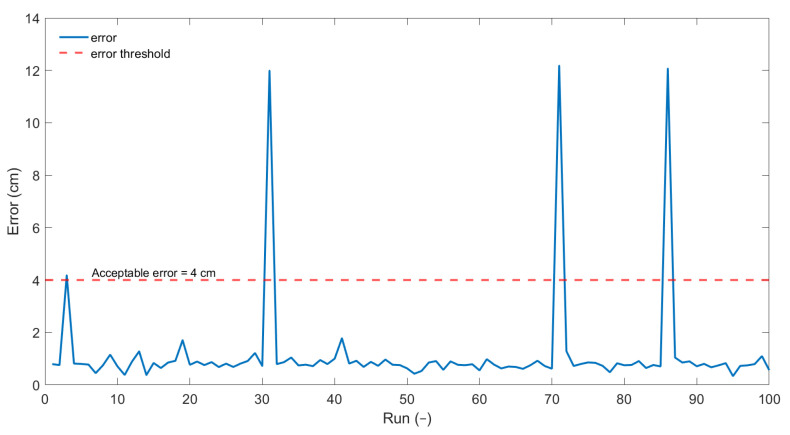
Error in damage localization for damage 7 for 100 PSO runs.

**Figure 10 sensors-22-06000-f010:**
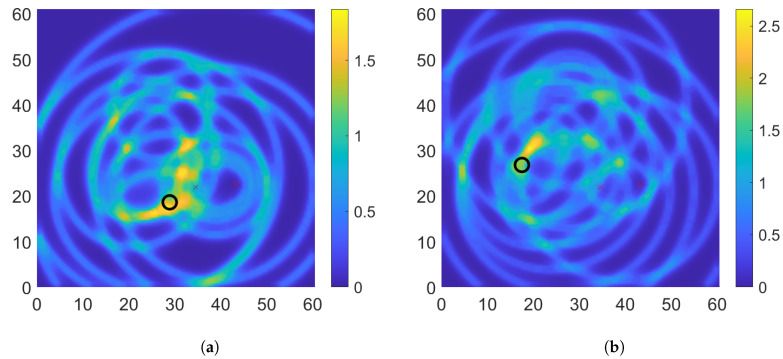
Damage maps with brute force method (**a**) D1 (**b**) D4.

**Figure 11 sensors-22-06000-f011:**
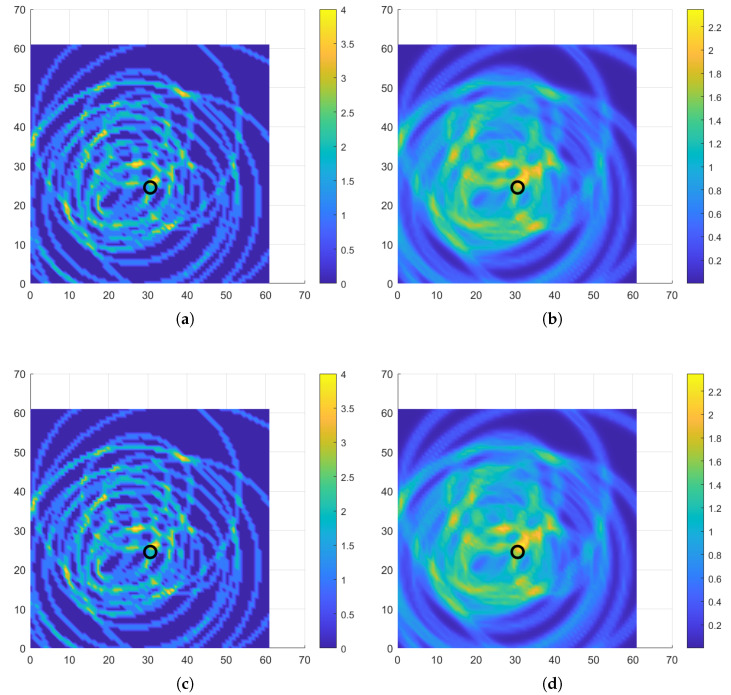
Damage maps comparing the two cost functions (**a**) Number of ellipses for D2 (**b**) Exponential function for D2 (**c**) Number of ellipses for D3 (**d**) Exponential function for D3.

**Table 1 sensors-22-06000-t001:** Comparison of the actual damage location and identified damage location.

Scenario	Actual Damage Location	Identified Damage Location	Error
	[cm]	[cm]	[cm]
D1	(28.9, 18.6)	(27.91, 17.45)	1.51
D2	(30.6, 24.5)	(31.64, 26)	1.82
D3	(26, 30)	(26.86, 31.32)	1.57
D4	(17.5, 26.8)	(18.79, 29.58)	3.06
D5	(41.4, 28.7)	(41.33, 29.48)	0.78
D6	(35, 36.8)	(35.76, 36.85)	0.76
D7	(22.7, 41.2)	(20.09, 41.66)	2.65

## Data Availability

The data presented in this study are available on request from the corresponding author. The data are not publicly available as it is part of ongoing research work.

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
