# Peer review of "Particle Swarm Optimization Algorithm for Guided Waves Based Damage Localization Using Fiber Bragg Grating Sensors in Remote Configuration"

_sensors, 2022, doi:10.3390/s22166000_

Round 1

Reviewer 1 Report

Comments:

This paper provides a proof of concept of the use of FBG sensors in remote configuration for the localization of damage in an aluminum plate through GW-based SHM. It is the first time to use remote bonded FBG sensor for damage localization. The paper is well organized and in a good written form with fluent English editing. However, this paper needs to be further revised based on the reviewer’s suggestion. Current version may be not suitable to be published.

1.       Line 4, repeated word of sensors should be deleted.

2.       Line 6, please give the full expression of GW.

3.       Line 36, what is WDM representing for?

4.       Line 45, the sensor sensitivity should be a natural property of FBG that will not be changed. Please elaborate about this sentence.

5.       Line 166, why did you use weight as simulate damage? What kind of damage such as crack, delamination does it simulate?

6.       Line 168, can rare-earth magnet firmly attach on the surface of aluminum plate?

7.       It is better to briefly introduce the principle for PZT-based guide wave and how does Remote bonding FBG sensor capture the guide wave through a schematic diagram.

Reviewer 2 Report

This manuscript provided a proof-of concept for the use of remotely bonded FBG for damage localization to improve the computational efficiency with particle swarm optimization (PSO), which allowed a significant improvement in the computation time for real-time damage localization. This manuscript is well written, and can be accepted before the following revisions.

1.       Is the proposed damage localization method dependent on the load and damage location?

2.       What is the sensor placement method of this manuscript? Such as the optimization method and evaluation criterion. Recent optimal sensor placement can be referred, e.g. 10.1016/j.ymssp.2021.108386 and “A novel two-step strategy of non-probabilistic ……” and “Optimal sensor placement based on dynamic condensation ……”.

3.       How to choose the best number of FBG sensors.

4.       What is the stability and initial value sensitivity of the optimization algorithm? Recent optimization algorithms can be referred, e.g. “A novel uncertainty-oriented regularization ……” and “A synchronous placement and size-based ……”.

5.       The manuscript has many grammatical and spelling errors, please check and correct the full text. For instance in Abstract, Structural Health (health) monitoring (SHM). “while able to” should be “while be able to”. “sensors sensors”. “In order to improve the computational efficiency a particle swarm optimization (PSO) based algorithm is developed.” missed a comma. ……
